# Genetic Editing of *Tomato Golgi-Localized Nucleotide Sugar Transporter 1.1* Promotes Immunity Against *Phytophthora infestans*

**DOI:** 10.3390/genes16040470

**Published:** 2025-04-21

**Authors:** Peize He, Yanling Cai, Yanzi Wang, Zhiqing Wang, Yaqing Lyu, Tao Li, Xingtan Zhang, Shaoqun Zhou

**Affiliations:** 1College of Informatics, Huazhong Agricultural University, Wuhan 430070, China; peizehe@outlook.com (P.H.); zhangxingtan@caas.cn (X.Z.); 2Shenzhen Branch, Guangdong Laboratory of Lingnan Modern Agriculture, Key Laboratory of Synthetic Biology, Ministry of Agriculture and Rural Affairs, Agricultural Genomics Institute at Shenzhen, Chinese Academy of Agricultural Sciences, Shenzhen 518120, China; caiyanling@caas.cn (Y.C.); wangyanziii@outlook.com (Y.W.); wangzhiqing@caas.cn (Z.W.); lvyaqing@caas.cn (Y.L.); 3School of Life Sciences, Henan University, Kaifeng 475004, China; 4Shenzhen Research Institute of Henan University, Shenzhen 518000, China; 5Guangdong Key Laboratory for New Technology Research of Vegetables, Vegetable Research Institute, Guangdong Academy of Agricultural Sciences, Guangzhou 510640, China; litao@gdaas.cn

**Keywords:** late blight, Golgi-localized nucleotide sugar transporter, susceptibility gene, *Solanum lycopersicum*, *Phytophthora infestans*

## Abstract

Background: Functional alleles of host plant susceptibility genes (S genes) can exacerbate the severity of diseases by enhancing pathogen compatibility. Genetic editing of the targeted host S genes has demonstrated remarkable efficacy in conferring broad-spectrum resistance across multiple crop species. We have previously identified a *Golgi-localized Nucleotide Sugar Transporter 1* homolog (*SlGONST1.1*) in the host plant *Solanum lycopersicum* as a susceptibility gene towards late blight caused by *Phytophthora infestans*. Methods: In this study, we performed a detailed characterization of tissue-specific and *P. infestans*-inducible expression patterns of this gene, and the subcellular localization of its encoded protein product. Results: Similar to phenotypes of two reported *Slgonst1.1* edited lines, two newly generated genetically edited lines of *SlGONST1.1* demonstrated enhanced resistance against *P. infestans* without obvious growth and developmental abnormality. Phytohormonal quantifications and reactive oxygen species measurements showed that an *Slgonst1.1* line had lower constitutive abscisic acid contents and depleted reactive oxygen species burst induced by pathogen-associated molecular pattern. Further comparative transcriptomic analyses revealed that the expression of defense-related genes is disproportionally up-regulated in the *Slgonst1.1* line. Conclusions: In summary, our findings confirmed *SlGONST1.1* as a functional host susceptibility gene towards late blight and shed light on the potential molecular mechanism underlying its function.

## 1. Introduction

Late blight (LB) caused by *Phytophthora infestans* de Bary is one of the most destructive diseases of tomato (*Solanum lycopersicum*) and potato (*Solanum tuberosum*) worldwide, causing up to 10 billion USD worth of yield loss and management cost annually [1]. The development of a genetically resistant crop cultivar has been recognized as an essential means for the sustainable management of LB [2].

Genetic resistance against LB has been primarily attributed to dominant *Resistance* (*R*) genes, which encode cytoplasmic immune receptors for pathogen-secreted effectors [2]. However, the rapid evolution of the pathogen populations in the field can quickly overcome the effective but strain-specific resistances conferred by these genes [3]. To the opposite of *R* genes, some host genes have been demonstrated to render the host plants more susceptible to diseases and, hence, are known as susceptibility genes [4]. These host susceptibility genes encode diverse protein products and are involved in different perspectives of host–pathogen interactions. For example, loss-of-function mutations in tomato *DMR6-1*, which encodes a salicylic acid (SA) deactivation enzyme, confers resistance against *P. capsici* and diverse phytopathogens [5]. Genetic editing of *DMR6-1* ortholog in potato also results in enhanced resistance against LB [6]. Other demonstrated susceptibility genes in potato that promote LB include *StSR4, StDND1,* and *StDMR1* [7,8,9]. Protein products of these genes are involved in calcium signaling (SR4), ionic transport (DND1), and phosphorrelay signaling (DMR1). Compared to *R* genes, however, functionally characterized susceptibility genes remain rare.

GOLGI LOCALIZED NUCLEOTIDE SUGAR TRANSPORTERs (GONSTs) are nucleotide sugar transporters that are critical for the glycosylation of membrane sphingolipids [10]. In Arabidopsis, *AtGONST1* has been demonstrated to be a broad-spectrum susceptibility gene, as genetic knock-out of *AtGONST1* can lead to constitutively heightened immunity but dwarfed growth [11,12]. The *A. thaliana* Columbia-0 genome contains five characterized GONST paralogs [13]. Among them, *AtGONST1* and *AtGONST2* show >50% similarity in their peptide sequences and have been shown to be functionally homologous [11,14,15]. Double knock-out of *AtGONST1* and *AtGONST2* could further elevate the constitutive levels of salicylic acid (SA) and reactive oxygen species (ROS) compared to *Atgonst1* single mutant plants [15]. Three additional *AtGONST* paralogs were predicted based on sequence homology, two of which have been functionally characterized as substrate-specific GDP-sugar transporters [13,14,16].

In a recently published post-inoculation time-course transcriptomics study, we identified a *P. infestans*-inducible *SlGONST1.1* homolog (*Solyc11T001957.3*, named *SlGONST1.1*) as a functional susceptibility gene that exacerbated LB [17]. In this study, we further characterized the molecular features of this gene and its protein product. Phenotypic characterization of two additional CRISPR–Cas9 edited lines targeting this gene (ATAGTAGCCAAGGCGGGGATAGG and CCAGCTATGACTTTAATGCGGGG) confirmed that knock-out of *SlGONST1.1* could enhance LB resistance without causing obvious growth and developmental defect. Further examination of the *Slgonst1.1* line revealed that this genetic perturbation led to reduced constitutive abscisic acid contents and elevated expression of defense-related genes, shedding light on the potential molecular mechanism underlying the immunity-suppressing function of *SlGONST1.1*.

## 2. Materials and Methods

### 2.1. Plant Materials and Growth Conditions

*Solanum. lycopersicum* cv. Micro-Tom (Lab strain) and CRISPR–Cas9 edited lines were used in this study. The growth conditions of plants and tomato transformation experiments are shown in references [17,18].

### 2.2. Plasmid Constructs

For the gene clone, the complete coding sequence of *SlGONST1.1* was inserted into pCE2 TA/Blunt-Zero Vector (Vazyme, C601-01) (Appendix A) to screen the positive clone and sequencing. The right single clone was used as a template to construct the *pSUPER::SlGONST1.1-EGFP* vector within *SpeI* and *KpnI* sites (Appendix A). The PCR program was as follows: 95 °C for 3 min, followed by 38 cycles of 95 °C for 20 s, 55 °C for 20 s, and 72 °C for 70 s. For CRISPR, two sgRNAs sites (ATAGTAGCCAAGGCGGGGATAGG and CCAGCTATGACTTTAATGCGGGG) were chosen for editing and cloned into the *pHSE401* plasmid containing the Cas9 coding sequence within the *BsaI* site (Appendix A). The specific steps are shown in Reference [17]. Genomic DNA was extracted from tentative mutant lines, and wildtype was used as a template to amplify the DNA fragment. The PCR products were sequenced and compared by SnapGene v 4.3. Primers are listed in Appendix A.

### 2.3. Phylogenetic Analysis

Phylogenetic analysis was based on *S. lycopersicum* and *A. thaliana* GONST homologs. The phylogenetic tree was constructed using MEGA v10.4.2 software based on the Maximum Likelihood model. The numbers in branches indicate the bootstrap values from 1000 replicates.

### 2.4. Subcellular Localization

*Agrobacterium tumefaciens* strain GV3101 harboring the binary vector for over-expression vector *pSUPER::SlGONST1.1-EGFP*, empty vector *pSUPER::EGFP*, Golgi marker vector *p35S::AtSYP61-mRFP*, and plasma membrane marker *p35S::PIP2-mRFP* [19,20] were grown in LB medium with antibiotics at 28 °C overnight. Bacteria were re-suspended in an infiltration buffer containing 10 mM MES, 10 mM MgCl_2_, and 150 μM acetosyringone in OD600 of 0.5-1. The bacteria harboring *pSUPER::SlGONST1.1-EGFP* was mixed with *p35S::AtSYP61-mRFP* and *p35S::PIP2-mRFP*. The bacteria harboring the empty vector did the same. The mixed bacteria were kept in the dark for two to three hours. Tobacco with four to five true leaves was used for injecting. After 48–60 h, leaves were detached for fluorescence observation by using a Leica SP8 confocal microscope equipped with 514 nm and 488 nm laser.

### 2.5. Quantitative RT-PCR

A plant RNA extraction kit (Huayueyang, Beijing, China) was used to extract total RNA from fresh leaf tissues following the manufacturer’s recommendations. cDNA was synthesized by using a HiScript II 1st Strand cDNA synthesis kit (+gDNA wiper) (Vazyme). qPCR product was obtained by ChamQ SYBR qPCR Master Mix (High ROX Premixed) kit (Vazyme Biotech Co., Ltd., Nanjing, China). Three biological and three technical replicates and the 2-^ΔΔCt^ method were used to evaluate all qPCR data (Bio-Rad Laboratories, Inc., Shanghai, China). The expression levels normalized to that of the housekeeping gene. All of the qPCR primers are listed in Appendix A.

### 2.6. P. infestans Infection Assays

Rye agar media plates were used to culture *P. infestans* strain T30-4 and 1306 (provided by Prof. Suomeng Dong from Nanjing Agricultural University). After 14–20 days, sporangia were collected in sterilized water, and their density was adjusted to 10,000 per mL for infecting; 10 μL of sporangia suspension was dropped on the underside of the leaves. The leaves were placed in a clean box, and a relative humidity of 90–100% was maintained. For 5–7 days, leaf infections were recorded, as well as the lesion diameter and lesion area [21].

### 2.7. Phytohormone Quantification

To analyze ABA, SA, IAA, JA, and JA-Ile in plant tissue, ten replicates of leaf samples were collected to test the phytohormones. Leaves for all time points were detached simultaneously and settled in moisturized Petri dishes for 6 h to minimize the impact of petiole cutting before the first inoculation. To avoid the impact of diurnal rhythms, samples were inoculated at different time points and collected simultaneously. Each biological replicate contained approximately 150 mg of leaves and was flash-frozen in liquid nitrogen. Before extraction, leaf samples were grinded to a fine powder before defrosting. Then, 1 mL of ethyl acetate, spiked with 20 ng of D6-ABA, D4-SA, D6-JA, and D5-JA-Ile as internal standards, was added. Supernatants were transferred to sterilized 2 mL microcentrifuge tubes after centrifugation at 10,000× *g* for 10 min at 4 °C; 500 μL ethyl acetate was added in each pellet and centrifuged instantaneously. Dryness in a vacuum concentrator (Eppendorf AG, Hamburg, Germany) was used to evaporate the supernatants; 500 μL of 70% methanol (*v*/*v*) was added to the residue and transiently centrifuged to clarify phases. Glass vials were used to collect the supernatants and then analyzed by HPLC-MS/MS (LCMS-8040, Shimadzu corporation, Tokyo, Japan).

An LC-20AD liquid chromatography system (Shimadzu corporation, Tokyo, Japan) was used to measure the characteristics of the metabolites, and 10 μL of each sample was injected into an ODS column (1.6 μm, 75 × 2 mm) (Shim-pack XR-ODS III, Shimadzu corporation, Tokyo, Japan) at a flow rate of 300 μL/min. Solvent A contained 0.05% formic acid and 5 mM ammonium formate, and solvent B only included methanol; they were used in a linear gradient at a mobile phase. The negative electrospray ionization mode was used to detect the parent ion with a specific mass-to-charge ratio and selected and fragmented to obtain its daughter ions. The corresponding compound’s chromatogram was generated from a specific daughter ion. The detailed steps of sample extraction and phytohormone measurements were carried out according to the description of reference [22].

### 2.8. ROS Measurements

A hole puncher was used to cut 0.125 cm^2^ leaf disks and incubated for 12 h in 96-well plates containing 200 μL water and kept in the dark in order to recover from wounding stress. The 200 μL solution containing luminol (20 μM), horseradish peroxidase (10 μg/mL) in sterile water, together with flg22 peptide (1 μM), was used to replace water as control. The luminescence was monitored continuously in 1 min intervals for 40 min with a Cytation5 (BioTek, Winooski, VT, USA) [23].

### 2.9. RNA-Seq Analysis

Clean reads were mapped to the *S. lycopersicum* ITAG4.1 genome by using the STAR software v2.4.0j [24]. Htseq v0.11.1 was applied to count the number of reads mapped to each gene. StringTie v1.3.3b software was used to calculate the FPKM (Fragments per kilobase per million) of each gene [25,26]. Differential expression genes (DEGs) between knock-out lines and wildtype were identified with the R package DESeq2 v1.22.2 (log2FoldChange > 1; FDR < 0.05; [27]). Gene ontology enrichment (GO) was obtained with the R packages GSEABase and GOstats.

### 2.10. Data Analysis

The data statistics in Figure 1, Figure 2, Figure 3, Figure 4 and Figure 5 were analyzed by One-Way ANOVA with Tukey‘s HSD (*p* < 0.05), and the data statistics in Figure 1, Figure 2, Figure 3, Figure 4 and Figure 5 were determined by Student’s *t*-test at ** *p* < 0.01 or * *p* < 0.05.

## 3. Results

### 3.1. Expansion and Molecular Divergence of SlGONSTs

Our previous time-course transcriptomics analyses of *S. lycopersicum, S. tuberosum,* and *N. benthamiana* revealed more than three thousand orthogroups that contained *P. infestans*-inducible homologs in the *Solanum* species only, one of which encoded homologs of *AtGONSTs* [17]. A more detailed phylogenetic analysis shows that both *P. infestans*-induced *S. lycopersicum* paralogs are orthologous to *AtGONST1*, whereas the third paralog is grouped with *AtGONST2* (Figure 1a). Genetic editing of one of the *AtGONST1* orthologs (*Solyc11T001957.3*, named *SlGONST1.1*) has been shown to promote tomato resistance against LB [17]. The length of the coding sequence of *SlGONST1.1* is 1302 bp, and the number of its corresponding amino acids is 434 (Appendix A). When transiently expressed in *N. benthamiana* leaves through co-infiltration, the protein product of this gene showed co-localization signals with both the Golgi apparatus marker and the plasma membrane marker proteins (Figure 1b). Constitutively, *SlGONST1.1* showed relatively uniform expression across six different tissue types, except for a lower expression level in immature fruits (Figure 1c). In an extended time-course q-RT-PCR experiment, *SlGONST1.1* showed transient induction by *P. infestans* zoospore inoculation at 8 h post-inoculation (hpi) and a second, stronger peak of induction at 72 hpi (Figure 1d).

### 3.2. SlGONST1.1 Lowers S. lycopersicum Resistance Against Late Blight Without Affecting Plant Growth and Development

Two additional genetic edit lines targeting *SlGONST1.1* (named *Slgonst1.1-3* and *Slgonst1.1-4*) were generated with the CRISPR–Cas9 technology in the Micro-Tom genetic background, and both edited alleles entailed different frameshift deletions (Figure 2a). Compared with wildtype, *Slgonst1.1-3* has 1bp insertion and 2 bp deletion in the first gRNA and 4bp deletion in the second gRNA. *Slgonst1.1-4* has 1bp deletion in the first gRNA and 4bp deletion in the second gRNA. Consistent with our previous observations, all four independent *SlGONST1.1* edited lines showed smaller lesion sizes and lower *P. infestans* biomasses compared to wildtype Micro-Tom plants, and there was no significant difference among these edited lines (Figure 2b–d). These results demonstrate that *SlGONST1.1* indeed functions as a host susceptibility gene that facilitates *P. infestans* infection of *S. lycopersicum* leaflet tissues.

Further phenotypic characterization of these *Slgonst1.1* mutant lines showed that knock-out of *SlGONST1.1* did not affect the plant height at maturity in the Micro-Tom genetic background (Figure 3a,b) Similarly, the fruit set per plant and the seed production statistics did not show significant difference between the mutant lines and their wildtype progenitor (Figure 3c–f). These results indicate that *SlGONST1.1* is not required for the normal growth and development of tomato plants, at least in this particular genetic background.

### 3.3. SlGONST1.1 Maintains Constitutive Abscisic Acid Levels and Suppresses PAMP-Induced ROS Burst

Since the *Slgonst1.1* mutants stably exhibit enhanced LB resistance, we set out to examine the physiological mechanisms underlying this phenotype. In Arabidopsis, both *AtGONST1* and *AtGONST2* additively suppress constitutive contents of SA and ROS [11,15]. However, we found no significant difference in SA contents between the two independent mutant lines and the wildtype control (Figure 4a). Instead, constitutive contents of abscisic acid (ABA) were consistently lowered in both *Slgonst1.1-1* and *Slgonst1.1-3* compared to wildtype (Figure 4b). In the same LC-MS analyses, contents of jasmonic acid, jasmonyl-isoleucine, and indole-3-acetic acid were also quantified in the same plant extracts, and no difference among these three genotypes was observed for any of these phytohormones (Figure 4c–e). We also assessed the ROS dynamics of the leaf discs of the two mutants and the wildtype control in response to flagellin-22 (flg22), a classic pathogen-associated molecular pattern (PAMPs). In both *Slgonst1.1-1* and *Slgonst1.1-3*, flg22 elicited stronger ROS bursts compared to wildtype, demonstrating that *SlGONST1.1* is a suppressor of PAMP-induced ROS burst (Figure 4f,g). In summary, these results suggested that *SlGONST1.1* may influence tomato resistance against LB by affecting constitutive ABA and PAMP-induced ROS metabolisms.

### 3.4. SlGONST1.1 Suppresses the Expression of Defense-Related Genes

To further investigate the influence of *SlGONST1.1* suggested by the physiological characterization of the *Slgonst1.1* mutants, we collected constitutive leaf transcriptomic data from the *Slgonst1.1-3* and the wildtype Micro-Tom plants (Appendix A). The principal component analysis (PCA) result of the RNA-Seq datasets showed clear distinctions between these two genotypes, revealing that the genetic perturbation of *SlGONST1.1* has an important impact on the overall gene expression profile in leaves (Figure 4a). More specifically, 1571 genes were found constitutively up-regulated in the *Slgonst1.1-3* mutant, whereas 813 genes were expressed at a higher constitutive level in wildtype (Figure 4b). Functional enrichment analyses of the up-regulated genes in the *Slgonst1.1-3* mutant revealed that a number of Gene Ontology (GO) terms typically associated with plant immunity, including defense response to fungus (GO: 0050832), regulation of peptidase activity (GO: 0052547), and phenylpropanoid metabolism (GO: 0009698), were over-represented among these genes (Figure 4c,d; Appendix A). These differential expression patterns were further confirmed with independent q-RT-PCR measurements of selected marker genes (Figure 4e). This comparative transcriptomic analysis revealed that *SlGONST1.1* could have an overarching suppressive effect on the transcription of diverse defense-related genes, which may explain its function as a host susceptibility gene.

## 4. Discussion

Host susceptibility genes can promote plant diseases through diverse molecular mechanisms such as facilitation of pathogen invasion, suppression of plant immunity, and nutritional support for pathogen growth [4]. In Arabidopsis, *Atgonst1* and *Atgonst2* have been shown to have enhanced constitutive contents of SA and ROS, which provide a mechanistic explanation for the auto-immune phenotypes observed in these mutants [11,15]. *S. lycopersicum* has two orthologs of *AtGONST1* (Figure 1a). We have shown that one of these orthologs, *SlGONST1.1*, exhibits *P. infestans*-induced transcription and can promote LB severity (Figure 1c and Figure 2) [17]. Unlike the reported *Atgonst1* mutants, the *Slgonst1.1* mutants did not show a significant difference in constitutive SA contents or ROS contents from the wildtype control (Figure 4). Instead, we observed lowered constitutive ABA levels and enhanced flg22-induced ROS bursts in the *Slgonst1.1* mutants (Figure 4). Neither of these phenotypes has been reported for any of the published *Atgonst* mutants. Collectively, this physiological evidence indicates divergent regulation of plant immunity between *SlGONST1* homologs and *AtGONST1.* Flagellin 22-induced ROS burst is a classic immune response marker, which has been previously adopted to illustrate the immunity-suppressing function of other genes involved in LB resistance [28,29]. It has also been reported that ABA accumulation can suppress plant immunity against LB [30,31]. Unfortunately, we do not find the other reported susceptibility genes, and ABA accumulation genes might interact with *SlGONST1.1* through yeast to hybrid screening. These functional consistencies suggest that the susceptibility-promoting function of *SlGONST1.1* may be mediated by positively regulating ABA accumulation while suppressing PAMP-induced ROS bursts.

Our comparative transcriptomic analysis of the wildtype and *Slgonst1.1-3* leaflet tissues revealed a pervasive suppressive effect on the expression of plant defense-related genes (Figure 5). Since *SlGONST1.1* is a Golgi- and membrane-localized transporter protein, it is unlikely that this protein can directly regulate gene transcription (Figure 1b). Rather, we postulate that the transcriptional regulatory effect of *SlGONST1.1* may be mediated by a downstream signaling cascade, which needs to be further elucidated. Noticeably, ABA-responsive genes have not been found to be overrepresented among the DEGs. This inconsistency between physiological characterizations and transcriptomic changes in the *Slgonst1.1* suggests that the influence of *SlGONST1.1* is potentially more complex than ABA-elicited responses alone. On the other hand, the lack of enrichment in ROS-related genes among the DEGs is expected since the ROS contents were only different between wildtype and the *Slgonst1.1-3* mutant upon flg22 elicitation, but not under the constitutive state.

At the molecular level, AtGONSTs are responsible for the transport of GDP-sugars, providing important substrates for the biosynthesis of glycosyl inositol phosphorylceramide (GIPC) sphingolipids [11,12]. In eudicots, membrane GIPCs could bind with necrosis and ethylene-inducing peptide 1-like (NLP) superfamily proteins, which are commonly produced as host-selective cytotoxins by phytopathogenic bacteria, fungi, and oomycetes [32]. The same study has further demonstrated that the characteristics of the sugar moieties on the GIPCs played critical roles in the host-selective molecular interactions between GIPCs and NLPs [32]. Hence, the conserved disease suppression phenotypes observed for *Atgonsts* and *Slgonst1.1* mutants may be attributable to perturbations to membrane GIPC profiles. However, further in vitro experiments are required to elucidate whether the GDP-sugar-transporting function is conserved for *SlGONST1.1* or not, and quantitative GIPC profiling of the *Slgonst1.1* mutants is necessary to determine the influence of these loss-of-function alleles in planta. Furthermore, how this GIPC modification activity of the GONSTs contributes to the defense-related physiological and transcriptional alterations observed in the *gonst* mutants also requires more detailed molecular investigation.

Besides the resistance phenotypes, we observed no significant difference in plant height at the flowering stage, the number of fruits produced per plant, and the seed set per fruit among the mutant lines and their wildtype progenitor (Figure 3). However, we recognize that the Micro-Tom background used in this study is naturally dwarfed and may mask some potential growth and developmental defects associated with these genetic manipulations.

In summary, our results confirm that *SlGONST1.1* is a host susceptibility gene towards LB and reveal that this function can be mediated by complex mechanisms that may involve ABA and ROS metabolism. These findings support that genetic editing of *SlGONST1.1* could be a viable approach to promote LB resistance in cultivated tomatoes and that the physiological influences of GONSTs have likely diverged between tomatoes and the model plant *A. thaliana.*

## Figures and Tables

**Figure 1 genes-16-00470-f001:**
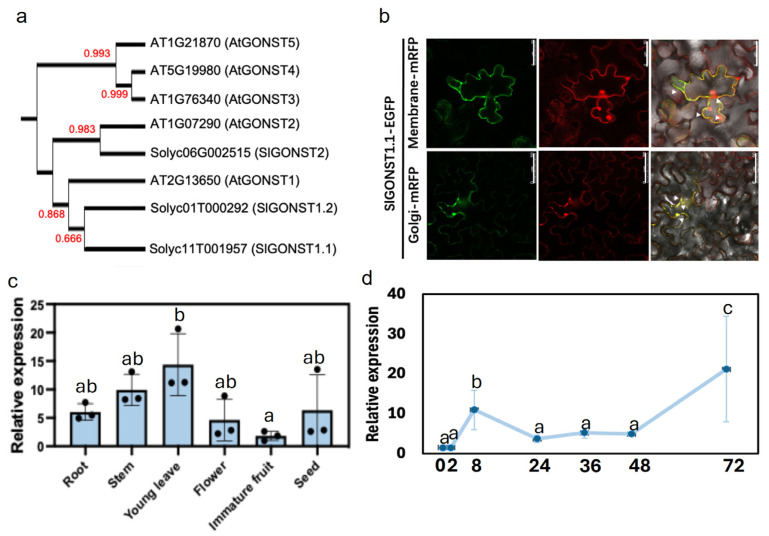
**Expansion and molecular divergence of SlGONSTs.** (**a**) Maximal likelihood phylogenetic relationship of *S. lycopersicum* and *A. thaliana* GONST homologs. Bootstrap support for each node is indicated in red. (**b**) Subcellular localization of SlGONST1.1. Coexpression of *SlGONST1.1-EGFP*/*p35S::AtSYP61-mRFP* (Golgi marker) and *SlGONST1.1-EGFP*/*p35S::PIP2-mRFP* (plasma membrane marker) in tobacco. The white arrow indicates the merged fluorescence of GFP fluorescence and mRFP fluorescence. Scale bars = 50 μM. Similar results were observed in at least three individual leaves. (**c**) The expression pattern of (**c**) *SlGONST1.1* in different tissues and organs of six-week-old plants. The black dots in (**c**) represent a biological repetition. (**d**) The expression pattern of *SlGONST1.1* after *P. infestans* zoospore inoculation at 0, 2, 8, 24, 36, 48, and 72 hpi (hour post inoculation). The blue dots in (**d**) represent the relative expression level of *SlGONST1.1*. The blue line is the trend line of relative expression level. Data in (**c**,**d**) are presented as mean ± sd (*n* = 3). Groups of significant differences were denoted by different letters (*p* < 0.05, one-way ANOVA followed by Tukey’s HSD). The lowercase letters in the (**c**,**d**) indicate whether there is a significant difference in the relative expression level of *SlGONST1.1* in one-way ANOVA analysis.

**Figure 2 genes-16-00470-f002:**
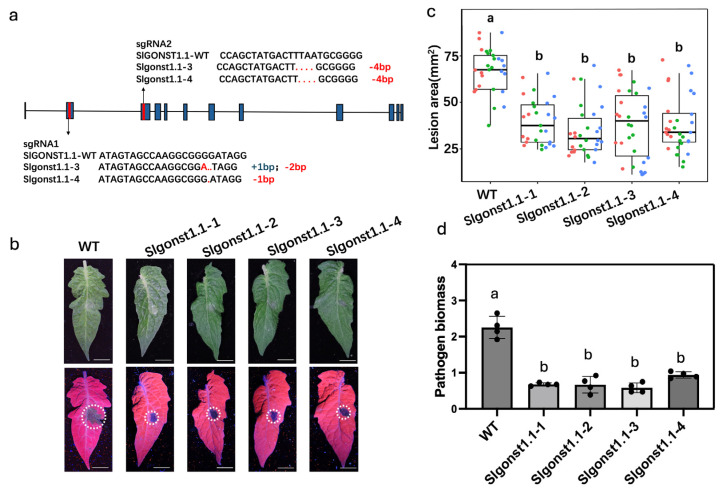
***SlGONST1.1* negatively affects plant resistance against *P. infestans*.** (**a**) Schematic representations of the CRISPR–Cas9-edited alleles of *SlGONST1.1*. The arrows indicate the gene editing of the corresponding exons of *SlGONST1.1*. (**b**) Leaves of wildtype and *Slgonst1.1* leaflets infected with *P. infestans* spore suspension under bright field (upper panels) or UV illumination (lower panels), respectively. Lesion areas outlined with white dotted lines. Scale bar = 6 mm. Photos taken at 6 days post-inoculation. (**c**) Lesion area of the wildtype and the *Slgonst1.1* knock-out lines. Red, green, and blue dots represent three biological replicates, and each point represents a sample. The whiskers (top and bottom) comprise values within 1.5 times the interquartile range (IQR). Center line, median; box limits, upper and lower quartiles. (**d**) Biomass of *P. infestans* in wildtype and the knock-out lines. Data are presented as means ± SD (*n* = 4). Each point represents an independent biological replicates. Significant differences in (**c**,**d**) determined according to one-way ANOVA analysis followed by Tukey’s test (*p* < 0.05). The lowercase letters in the (**c**,**d**) indicate whether there are significant differences between different groups.

**Figure 3 genes-16-00470-f003:**
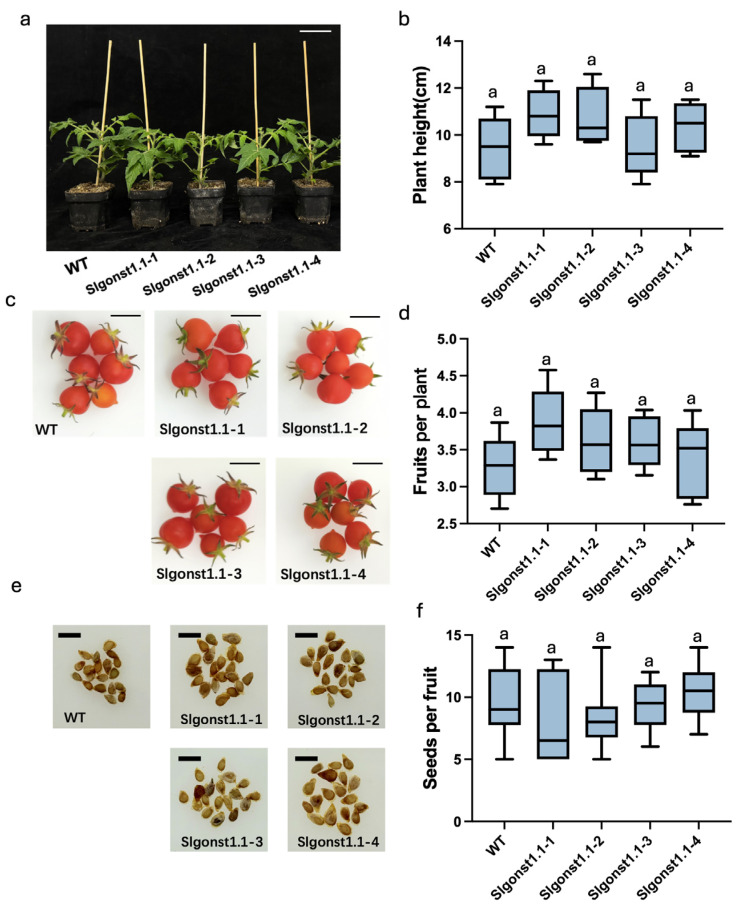
***Slgonst1.1* mutants show similar growth stature and reproductive traits to wildtype progenitors.** Representative photos (**a**) and plant height statistics (**b**) of six-week-old Micro-Tom and *Slgonst1.1* mutant plants. Scale bar = 4.5 cm. Representative photos (**c**) and fruit set per plant statistics (**d**) of twelve-week-old Micro-Tom and *Slgonst1.1* mutant plants. Scale bar = 1.25 cm. Representative photos (**e**) and seed set per fruit statistics (**f**) of twelve-week-old Micro-Tom and *Slgonst1.1* mutant plants. Scale bar = 3 mm. Data in (**b**,**d**,**f**) are presented as means ± SD (*n* = 10). Lowercase letters indicate significant differences according to one-way ANOVA analysis followed by Tukey’s test (*p* < 0.05). The lowercase letters in (**b**,**d**,**f**) indicate whether there are significant differences between different groups.

**Figure 4 genes-16-00470-f004:**
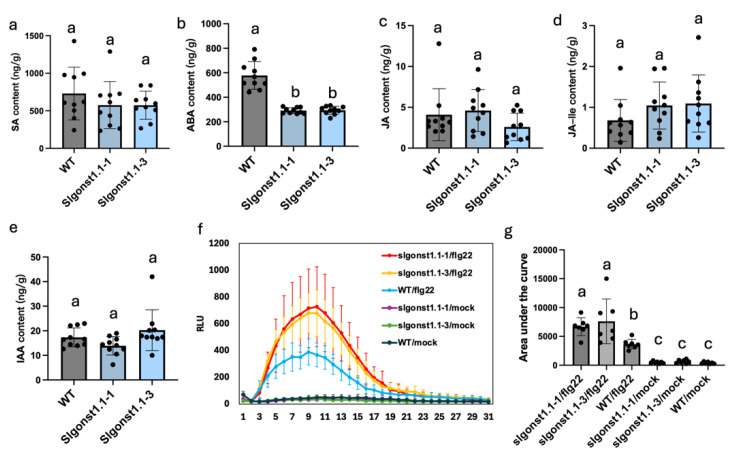
**Phytohormone contents and PAMP-induced ROS dynamics detection in *Slgonst1.1* knock-out lines.** Concentrations of (**a**) salicylic acid (SA), (**b**) abscisic acid (ABA), (**c**) jasmonic acid (JA), (**d**) jasmonyl isoleucine (JA-Ile), and (**e**) indole-3-acetic acid (IAA) in wildtype and *Slgonst1.1* knock-out lines. Statistical significance was assessed by one-way ANOVA followed by Tukey’s HSD (*n* = 10, *p* < 0.05). (**f**) Time-course quantification of reactive oxygen species in leaf discs of different genotypes elicited with flg22 (*n* = 7; data presented as mean ± sd). Sterile water was taken as mock treatment control. (**g**) Areas under curve calculated for the ROS quantification experiment. Genotypes of statistically significant differences are denoted by different letters (one-way ANOVA; *p* < 0.05). Each point represents an independent biological replicates. The lowercase letters indicate whether there are significant differences between different groups.

**Figure 5 genes-16-00470-f005:**
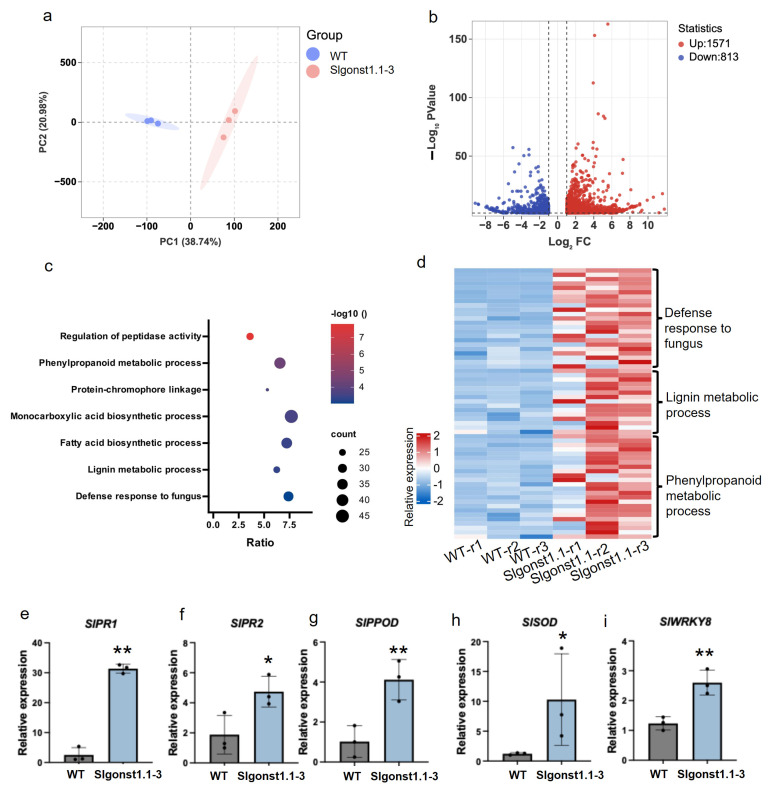
**Comparative transcriptomic analysis of *Slgonst1.1-3* and its wildtype progenitor.** (**a**) PCA of the transcriptomes of wildtype (WT) and *Slgonst1.1* tomato seedling leaflets. The colored ovals indicate 60% confidence intervals. The two dashed lines represent that the variance of PC1 and PC2 is 0, indicating that the data has no variability. (**b**) Differentially expressed genes between WT and *Slgonst1.1* leaflet depicted by a volcano plot. The horizontal dotted line represents *p* < 0.05. Two vertical dashed lines represent |Log_2_FC|≥1 (**c**) Top enriched GO term among the *Slgonst1.1*-upregulated DEGs. (**d**) Expression patterns of the *Slgonst1.1*-upregulated DEGs associated with the defense-related enriched GO terms. (**e**–**i**) qRT-PCR analysis of marker genes representative of the defense-related DEGs. Data are presented as mean ± sd (*n* = 3). Asterisks indicate a significant difference between transgenic plants and WT as determined by Student’s *t*-test at ** *p* < 0.01 or * *p* < 0.05.

## Data Availability

The cleaned reads of the RNA-seq dataset were deposited at the Sequence Read Archive (SRA) of the China National Center for Bioinformation under accession number PRJCA034969.

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
