# Peer review of "Genetic Editing of Tomato Golgi-Localized Nucleotide Sugar Transporter 1.1 Promotes Immunity Against Phytophthora infestans"

_genes, 2025, doi:10.3390/genes16040470_

Round 1
Reviewer 1 Report
Comments and Suggestions for Authors
In the title add the gene name "Genetic editing of SlGONST1.1 promotes immunity against Phytophthora infestans"
Abstract: The abstract currently mentions host plant susceptibility genes; however, it is necessary to specify the name of the host plant used in the study. Clearly stating the host plant will provide better context and clarity to the reader.
The section on plasmid constructs requires a more detailed explanation, including every minute detail about the components used in the construction process. This should include the specific vectors used, the cloning strategy implemented, and any verification steps undertaken to confirm successful construction. Providing this level of detail will ensure transparency and reproducibility of the methodology.
Regarding the cloned genes, it is essential to include information on their characteristics. The size of the coding sequence (CDS) should be specified, and the complete nucleotide sequence can be provided as supplementary information. Additionally, the corresponding amino acid sequence must also be included to facilitate a comprehensive understanding of the gene and its functional properties.
In Figure 1D, it is necessary to clearly define and label the x-axis to ensure that readers can interpret the data accurately. Figures 2A, 2B, and 2C require an increase in font size to enhance readability, making the text clearer and more accessible. In Figure 3, specifically panels A, C, and E, the image size should be increased to improve visualization and better illustrate the experimental findings. Furthermore, Figure 3 lacks statistical significance representation, which is a crucial component for validating the results. Including statistical significance will strengthen the credibility and interpretation of the findings.

Can be improved
Author Response
We would like to thank the reviewers for evaluating our revised manuscript. Please see the point-by-point responses below with references to this first round of revision made in the manuscript.
Comments1:In the title add the gene name "Genetic editing of SlGONST1.1 promotes immunity against Phytophthora infestans"
Response1:Thank you for pointing this out. The term “SlGONST1.1” is replaced as “tomato Golgi-localized Nucleotide Sugar Transporter. 1.1” in title.
Comments2:Abstract: The abstract currently mentions host plant susceptibility genes; however, it is necessary to specify the name of the host plant used in the study. Clearly stating the host plant will provide better context and clarity to the reader.
Response2: Thank you for pointing this out. The host plant is mentioned in line 21.
Comments3:The section on plasmid constructs requires a more detailed explanation, including every minute detail about the components used in the construction process. This should include the specific vectors used, the cloning strategy implemented, and any verification steps undertaken to confirm successful construction. Providing this level of detail will ensure transparency and reproducibility of the methodology.
Response3: Thank you for pointing this out. The detail information about the vector construction process is added in plasmid constructs. section in line 94-100.
Comments4:Regarding the cloned genes, it is essential to include information on their characteristics. The size of the coding sequence (CDS) should be specified, and the complete nucleotide sequence can be provided as supplementary information. Additionally, the corresponding amino acid sequence must also be included to facilitate a comprehensive understanding of the gene and its functional properties.In Figure 1D, it is necessary to clearly define and label the x-axis to ensure that readers can interpret the data accurately. Figures 2A, 2B, and 2C require an increase in font size to enhance readability, making the text clearer and more accessible. In Figure 3, specifically panels A, C, and E, the image size should be increased to improve visualization and better illustrate the experimental findings. Furthermore, Figure 3 lacks statistical significance representation, which is a crucial component for validating the results. Including statistical significance will strengthen the credibility and interpretation of the findings.
Response4: Thank you for pointing this out.
The coding sequence (CDS) and the corresponding amino acid sequence of SlGONST1.1 are adding to the Supplemental table 2 and to the Supplemental table 3. Their characteristics are described in line 188-189.
The x-axis of Figure 1D is defined in line 205-206.
The font size of Figures 2A, 2B, and 2C are changed to enhance readability.
The image size of Figure 3 A, C, and E are adjusted to improve visualization and better illustrate. the experimental findings.
The statistical significance is described in figure 3 legend in line 241-242.

Reviewer 2 Report
Comments and Suggestions for Authors
The current study has focused on characterizing the molecular features of SlGONST1.1 and its protein product, as well as examining the phenotypic edited lines against P. infestants. The results are of interest in improving the resistant ability of cultivated tomatoes for further developing sustainable agricultural production. However, the current form of the manuscript has not been well organized and documented. The authors can find some useful comments and suggestions below to improve the quality of the manuscript.
- The authors should explain the capital letters of “1” and “Slgonst1.1”;
- Reword the objectives of the current study for easier understanding
- In Materials and Methods, subsection “2.1. Plasmid constructs” needs to provide more information in detail, and should draw a Fig or Scheme of how to design and construct the vector;
- Should briefly describe the mutant and wild-type lines? Where are they from?
- Some information “… empty vector pSUPER::EGFP, Golgi marker vector p35S:: AtSYP61-mRFP and plasma membrane marker p35S::PIP2-mRFP “ on line 94-97 pages 2-3 have to provide more information;
- Should add a subsection like “ Plant materials” and narrate all plant materials used in this study, for example: in line 102 page 3, “Four to five-week-old tobacco was used for injecting”, tomato, mutant and wild plants etc., How did the author grow them and what conditions?
- Where did the author take the infestant strain T30-4 and 1306? Provide more information!
- Have the applied methods in this study been developed by the authors?, if so, please mention or the cited references needed for reliable protocols.
- How were the data in Figs 1, 2, 3, 4……calculated without statistical analyses? In my opinion, the subsection “Data analyses” should be added at the end of the Materials and Methods section;
- The authors mentioned "phenotypes of two Slgonst .1.1 edited lines" All information of these lines need to be provided in the Materials and Methods in details;
- Discussion should be improved; for example, the interaction of SlGONST1.1 with other susceptibility genes or pathways
- Check all spelling and grammar errors, cited references and list of references need to fit one another; for example the scientific name of late blight in the title is not correct, etc.

English needs some improvement for easier understanding
Author Response
I congratulate the authors for their work and manuscript.
Comments1:The authors should explain the capital letters of “1” and “Slgonst1.1”;
Response1:Thank you for pointing this out. In our previous paper, we found this gene which was the homolog of Arabidopsis Golgi-localized Nucleotide Sugar Transporter 1(AtGONST1). According to the phylogeny analysis, tomato has two homolog of AtGONST1, so we named the first reported gene SlGONST1.1 and the unreported denoted as SlGONST1.2(Cai et al., 2025).
Slgonst1.1 indicated the knock-out line of SlGONST1.1 which is described in line 188-189.
Cai Y., Wang Z., Wan W., et al. Time-course dual RNAseq analyses and gene identification during early stages of plant-Phytophthora infestans interactions. Plant physiology, 2025, 197(4): kiaf112.
Comments2:Reword the objectives of the current study for easier understanding
Response2:Thank you for pointing this out. The objectives of the current study has been revised as “Functional alledels of host plant susceptibility genes (S genes) can exacerbate the severity of diseases by enhancing pathogen compatibility. Genetic editing the targeted host S genes has been demonstrated to be remarkable efficacy in conferring broad-spectrum resistance across multiple crop species.” in Abstract section.
Comments3.In Materials and Methods, subsection “2.1. Plasmid constructs” needs to provide more information in detail, and should draw a Fig or Scheme of how to design and construct the vector;
Response3:Thank you for pointing this out. The detail information about the vector construction process is added in plasmid constructs section in line 94-100. The Scheme of the plasmid constructs are involved in Supplemental figure1-3.
Comments4.Should briefly describe the mutant and wild-type lines? Where are they from?
Response4:Thank you for pointing this out. The wild-type lines are kept by our lab and mutant lines was gained by vector construction and Agrobacterium-mediated tomato transformation was performed by Weimi Biotechnology Co. (Baige, Jiangsu) following an established protocol. This is described in line 87-89.
Comments5. Some information “… empty vector pSUPER::EGFP, Golgi marker vector p35S:: AtSYP61-mRFP and plasma membrane marker p35S::PIP2-mRFP “ on line 94-97 pages 2-3 have to provide more information;
Response5:Thank you for pointing this out. The original source of the vector p35S::AtSYP61-mRFP and p35S::PIP2-mRFP are added in the line 112.
Comments6. Should add a subsection like “ Plant materials” and narrate all plant materials used
in this study, for example: in line 102 page 3, “Four to five-week-old tobacco was
used for injecting”, tomato, mutant and wild plants etc., How did the author grow
them and what conditions?
Response6:Thank you for pointing this out. The subsection “Plant materials and growth conditions” has been added in line 82-89.
Comments7. Where did the author take the infestant strain T30-4 and 1306? Provide more
information!
Response7:Thank you for pointing this out. The source information of Phytophthora infestans strain T30-4 and 1306 has been added in Acknowledgement section.
Comments8. Have the applied methods in this study been developed by the authors?, if so, please
mention or the cited references needed for reliable protocols.
Response8:Thank you for pointing this out. Agrobacterium-mediated transformation of tomato References were added to line 89
The literature reference of Phytophthora infestans infection assays is in line 134.
The literature reference of ROS measurements is in line 166.
Comments9. How were the data in Figs 1, 2, 3, 4……calculated without statistical analyses? In
my opinion, the subsection “Data analyses” should be added at the end of the
Materials and Methods section;
Response9:Thank you for pointing this out. The subsection “Data analyses” has been added in line 175-178.
Comments10. The authors mentioned "phenotypes of two Slgonst .1.1 edited lines" All
information of these lines need to be provided in the Materials and Methods in
details;
Response10:Thank you for pointing this out. The information of two edited lines has been added in line 96-100 and 214-216.
Comments11. Discussion should be improved; for example, the interaction of SlGONST1.1 with
other susceptibility genes or pathways
Response11:Thank you for pointing this out. We had discussed the the homolog of SlGONST1.1 in Arabidopsis AtGONST1, whose mutant Atgonst1 enhanced constitutive contents of SA and ROS while the content of Slgonst1.1 knock-out line doesn’t affect the constitutive contents of SA which means SlGONST1.1 doesn’t enhance the resistance through SA pathways. Also, Slgonst1.1 knock-out line enhance ROS burst which is similar to Atgonst1 mutant. This part of discussion has been exhibited in line 306-314.
Although we showed the ABA content in Slgonst1.1 knock-out line decreased compared with wildtype, but we couldn’t find the interaction protein which connects with ABA pathway in other experiment. It has also been reported that ABA accumulation can suppress plant immunity against late blight. This makes us to think how SlGONST1.1 affect ABA pathway to enhance resistance in tomato for further research in line 314-320.
And about the interaction with other susceptibility genes, we don’t find the other susceptibility genes might interact with SlGONST1.1 through yeast two hybrid screening. We might screen the interacted protein through other method like IP-MS. This part of discussion has showed in line 316-318.
Comments12. Check all spelling and grammar errors, cited references and list of references need
to fit one another; for example, the scientific name of late blight in the title is not
correct, etc.
Response12:Thank you for pointing this out. We carefully checked all spelling and grammar errors cited references and list of references to improve paper quality.

Reviewer 3 Report
Comments and Suggestions for Authors
I congratulate the authors for their work and manuscript.
Manuscript ID: genes-3541466 "Genetic editing of SlGONST1.1 promotes immunity against Phythophthora infestans" by He and collaborators shows characterization of a susceptibility gene in tomato. The authors performed a careful sequence characterization to properly identify the orthologous group of each SlGONST1.1 paralog, and a detailed analysis of subcellular localization and temporal tissue-specific expression. The authors also determined the role of SIGONST1.1 in disease susceptibility and plant growth and development. Finally the authors also investigated the effect of this gene in phytohormone and ROS levels, and expression of defense-related genes. Taken together, the authors make a compelling case to classify this gene as a susceptibility factor to Phytophthora. The manuscript is concise and well written, although I do recommend a final verification by the editing team to adjust some minor language details. Methods provide important experimental information and Results are presented with statistical significance. The Discussion and Conclusions are supported by the data. I have no issues with the current version and will recommend minor revision to give the authors the opportunity to work on these minor suggestions below:
Line 54: Substitute "phosphor-relay" with "phosphorelay".
Line 84: Include gRNA sequences.
Line 149-150: Please clarify if the intended meaning was "flg22-peptide (Phytotech) was used replacing water as control".

Author Response
I congratulate the authors for their work and manuscript.
I have no issues with the current version and will recommend minor revision to give the authors the opportunity to work on these minor suggestions below:
Comments1:Line 54: Substitute "phosphor-relay" with "phosphorelay".
Response1:Thank you for pointing this out. The term "phosphor-relay" has been substituted as "phosphorelay" in line 55.
Comments2:Line 84: Include gRNA sequences.
Response2:Thank you for pointing this out. gRNA sequences are described line 96-97
Comments3:Line 149-150: Please clarify if the intended meaning was "flg22-peptide (Phytotech) was
used replacing water as control".
Response3:Thank you for pointing this out. The sentence has changed as “flg22-peptide (Phytotech) was used replacing water as control” in line 165
